# Multiple 17-OHP Cutoff Co-Variates Fail to Improve 21-Hydroxylase Deficiency Screening Accuracy

**DOI:** 10.3390/ijns8040057

**Published:** 2022-10-25

**Authors:** Preet K. Matharu, Patrice K. Held, David B. Allen

**Affiliations:** 1Department of Pediatrics, University of Wisconsin School of Medicine and Public Health, Madison, WI 53726, USA; 2Oregon State Public Health Laboratory, Oregon Health Authority, Hillsboro, OR 97124, USA; 3Department of Pediatrics, Division of Endocrinology and Diabetes, University of Wisconsin School of Medicine and Public Health, Madison, WI 53792, USA

**Keywords:** 21-hydroxylase deficiency, congenital adrenal hyperplasia, 17-hydroxyprogesterone, birth weight, collection time, newborn screening

## Abstract

To improve the positive predictive value (PPV) of newborn screening for 21-hydroxylase deficiency (21OHD), co-variates have been used to modify 17-hydroxyprogesterone (17OHP) cutoffs. The objective of this study is to evaluate whether 17OHP screening cutoffs adjusted for both collection time (CT) and birth weight (BW) improved the sensitivity and PPV of 21OHD screening. Unaffected newborn screening samples were stratified based on BW and CT to establish 17OHP concentration cutoffs at the 95th and 99th percentile. These cutoffs were applied to a cohort of confirmed cases of 21OHD to determine the sensitivity and PPV of the modified screening parameters. 17OHP cutoffs at the 99th percentile, adjusted for BW and CT, had a sensitivity of 96.3% and a specificity of 98.9%, but a relatively low PPV (0.130) for the identification of 21OHD and did not detect all cases. Use of the 95th percentile further increased sensitivity to 98.1% but resulted in a notably lower PPV (0.027). Alternative approaches that do not rely exclusively on 17OHP are needed to improve newborn screening accuracy for 21OHD.

## 1. Introduction

Congenital adrenal hyperplasia (CAH) is a group of autosomal recessive disorders characterized by defects in the cortisol synthesis pathway, with the majority of cases due to impaired function of the 21-hydroxylase enzyme. The severity of the enzyme deficiency is varied, leading to a spectrum of clinical presentations that has been grouped into two forms: classic and non-classic 21-hydroxylase deficiency (21OHD). If untreated, infants with the severe classic form develop salt-wasting with hyponatremia, hypokalemia, and life-threatening adrenal crisis. Additionally, in females, there is genital virilization of varying degrees due to in utero exposure to androgens [1].

Screening for 21OHD was first implemented in 1978 using a radioimmunoassay to measure 17-hydroxyprogesterone (17OHP) concentrations. Today, screening laboratories continue to measure 17OHP concentrations using a dissociation-enhanced lanthanide fluorescence immunoassay in automated systems (DELFIA). Screening for 21OHD is challenging because the positive predictive value (PPV) of the immunoassay is low (less than 10%) [2]. Reasons for a low PPV include elevated 17OHP levels in newborns (especially premature infants) shortly after delivery that are not associated with a diagnosis of 21OHD, immaturity of the adrenal enzymes within the steroidogenic pathway, and a heightened stress response in newborns with severe illness. Additionally, slight differences in laboratory protocols for measuring 17OHP concentrations and the potential for cross-reactivity with other steroids has led to false positive results [3,4,5,6,7].

Current screening practices have also demonstrated a propensity for missing cases of 21OHD, with one Minnesota-based study reporting a false negative rate of 22% [8]. Studies have identified reasons for false negative results, including early collection times that do not account for late-rising 17OHP levels in newborns, immaturity of adrenal steroid production in premature newborns, and exposure to maternal cortisol or glucocorticoid treatments [2,4]. Other contributing factors include the use of less sensitive screening thresholds in which 17OHP concentrations in affected newborns fall just below established cutoffs.

To improve the screening accuracy for 21OHD, multiple co-variates, including birth weight (BW), gestational age, and age at time of specimen collection, have been assessed to determine which parameter would be most appropriate for establishing a more sensitive 17OHP screening cutoff [5,6,7,9,10]. While these approaches have modestly improved screening sensitivity, the PPV of screening remains low.

To further improve sensitivity and the PPV of current screening practices, we utilized collection time (CT) in conjunction with BW to determine if 17OHP screening cutoffs can be optimized for the detection of 21OHD cases. We acknowledge that some studies have reported an improved correlation of gestational age with 17OHP levels, as it is more directly associated with HPA axis development [10]. However, the decision was made to use BW because estimation of gestational age is less accurate given variable dating methods; several other studies have also proven that birth weight is a reliable co-variate to use for the adjustment of 17OHP levels, with a significant improvement in false positive rates for screening [6,7].

## 2. Materials and Methods

### 2.1. Specimens

This study involved a retrospective assessment of 17OHP concentrations measured in newborn screening specimens (dried blood spots) received by the Wisconsin State Laboratory of Hygiene for routine 21OHD screening. The specimens were collected from unaffected newborns born between 1 January 2019 and 30 June 2020 (n = 104,080). Each sample was associated with specific data parameters pertaining to sex, CT, gestational age, BW, initial or repeat sample, and 17OHP serum concentration. Specimens were excluded from the dataset if any of these parameters were missing or undefined, or if the collection method was deemed unsatisfactory. Specimens from unaffected newborns were separated into two cohorts: one cohort of specimens collected between 1 January 2019 and 31 December 2019 (n = 69,798) that was used to establish 17OHP reference ranges at the 95th and 99th percentiles, and a second cohort of specimens collected between 1 January 2020 to 30 June 2020 (n = 34,282) that was used to compare and validate the modified 17OHP cutoffs within the normal population.

A third cohort of confirmed cases (n = 55) of 21OHD identified in Wisconsin and several other states were used to test the sensitivity and PPV of the modified 17OHP screening cutoffs. For these confirmed cases, data pertaining to BW, age at collection, sex, and serum 17OHP level based on immunoassay testing were used in the analysis. Stratification of cases by severity of disease, simple virilizing or salt wasting phenotypes, could not be performed due to reported discrepancies in confirmed cases and lack of confirmatory molecular data. Therefore, our pool of 55 confirmed cases was assumed to contain both phenotypes.

### 2.2. Screening Assay

Whole blood specimens were collected from newborns on Whatman GmbH grade 903 filter paper and sent to the laboratory. The concentration of 17OHP was measured using the dissociation-enhanced lanthanide fluorescence immunoassay in automated systems according to the manufacturer’s protocols (PerkinElmer Life and Analytical Sciences protocols, Shelton, CT, USA) [2].

### 2.3. Statistical Analysis

17OHP concentration levels collected from unaffected newborns (first cohort, n = 69,798) were grouped based on the specimens’ CT and BW. BW data were separated into the following categories: <1000 g, 1001–1500 g, 1501–2500 g, and >2500 g. These categories were based on previous studies implementing this BW co-variate, as well as World Health Organization classifications for pre-term and term delivery birth weights. CT was separated into the following categories: ≤23 h, 24–72 h, 73 h–18 days, 19–45 days, and >45 days. These categories were established to account for repeat newborn samples collected outside of the standard 24–48 h time period. It should be noted that although states mandate that newborn screening samples be collected between 24–48 h to account for false elevations with prematurely collected samples, examination of 17OHP levels from unaffected newborns collected between 24–48 h and 49–72 h across our established BW categories did not yield significant differences between the two collection times within our datasets. 

Mean and median 17OHP concentrations were calculated to confirm appropriate grouping of these established categories. Within each of these categories, the 95th and 99th percentiles were calculated and designated as modified 17OHP concentration cutoffs for screening. These percentiles were then applied to a second cohort of unaffected newborns and a cohort of confirmed cases of 21OHD to assess the sensitivity, specificity and PPV of new proposed thresholds. Data were analyzed using Microsoft Excel 2016 and GraphPad Prism Version 9.1.0 (GraphPad Software, San Diego, CA, USA) software programs.

## 3. Results

### 3.1. Distribution of 17OHP in the Unaffected Population

The impact of CT and BW on the distribution of 17OHP concentrations within the unaffected newborn population was evaluated. Within the unaffected cohort of specimens, an inverse correlation between BW and 17OHP concentrations was observed. Babies with a BW <1500 g had higher 17OHP concentrations than babies with a BW greater than 1500 g. Regarding CT, for infants <1000 g, a peak 17OHP concentration was observed for the collection time of 24–72 h. For all other BW categories, an inverse correlation between CT and 17OHP concentrations was noted, with earlier collection times yielding higher concentrations of 17OHP. Figure 1 demonstrates the stratified distribution patterns of 17OHP concentrations for unaffected newborn screening samples based on BW and CT.

### 3.2. Development and Application of 17OHP Concentration Cutoffs

Reference ranges set at the 95th and 99th percentiles were established for the unaffected population (specimens received between 1 January 2019 and 31 December 2019) based on the stratified BW and CT categories. These ranges were then compared to the second cohort of newborns (collected between 1 January 2020 and 30 June 2020) to ensure the appropriate stratification of categories and percentiles. These modified percentile ranges were also applied to the cohort of 55 confirmed cases of 21OHD in order to determine the sensitivity, specificity, and PPV of the modified cutoffs.

Application of the 99th percentile demonstrated a sensitivity of 96.2% and a specificity of 98.9%, with a calculated PPV of 0.130. Using a 95th percentile cutoff, sensitivity further increased to 98.1%. However, specificity decreased to 94.3% with a markedly lower PPV (0.027). Notably, the 95th percentile cutoff still failed to detect all cases of 21OHD. It was determined that the cutoff would need to be lowered to the 93rd percentile to detect all cases in our given cohort. Table 1 summarizes the results for sensitivity, specificity, and PPV based on the modified 95th and 99th percentiles.

## 4. Discussion

This retrospective analysis examined whether the combined use of BW and CT covariates to modify 17OHP concentration cutoffs would improve the sensitivity and PPV of 21OHD screening. The results showed that the application of the modified 17OHP cutoffs at the 99th percentile led to a relatively high sensitivity of 96.3% and a notably higher PPV of 0.130 (improved in comparison to the reported average PPV) [2,11,12,13]. Application of the 95th percentile, while further improving sensitivity (98.1%), lowered both specificity (94.3%) and the PPV (0.027). 

It must be noted that our study utilized a large cohort of newborns (34,282) that were presumptively unaffected with 21OHD as the control group. Some of these newborns may have 21OHD but this had not yet come to clinical attention at the time of publication (false negatives). Therefore, the sensitivity, specificity and PPV as defined above should be interpreted with caution. 

Previous studies have produced similar screening sensitivities of greater than 90% upon the modification of 17OHP cutoffs based on multiple co-variates [11,14]. These findings indicate that establishing 17OHP cutoffs at a lower percentile, while improving sensitivity and reducing false negative rates, introduces additional false positive cases and detracts from the PPV of current 21OHD screening assays that rely on 17OHP as a primary screening marker.

The persistence of a low PPV in screening for 21OHD suggests that current screening practices must be modified. Reported modifications have included the implementation of a mandatory repeat screen several days after birth, which improves the sensitivity and PPV of screening [12]. Additionally, Wisconsin and other states have added an expanded steroid profile panel, which includes a variety of steroidogenic molecules as a second-tier test for newborn screening. The use of these second-tier tests, including those involving liquid chromatography with tandem mass spectrometry, can improve false positive rates and increase PPV up to 11% compared to the use of immunoassays alone [2,13,15,16,17,18]. It should be noted however, that while second-tier testing has improved the PPV, this screening method comes with several limitations. Analysis time is notably longer with an expanded steroid profile, impacting the time from specimen receipt to reporting of critical results. Additionally, the use of lower thresholds for 17OHP to detect all confirmed cases (projected 93rd percentile in this study) results in an exponential increase in the number of specimens (greater than five-fold) requiring second-tier testing, directly impacting the screening laboratory workload. While it is evident that second-tier testing is beneficial for confirming or negating false positive cases, further evaluation is needed to determine the impact of an expanded steroid profile as a screening tool, particularly with regard to laboratory workflow, timeliness, and cost-effectiveness. 

Additional steroidogenic molecular markers have also been proposed as initial screening tools for the detection of 21OHD. For example, the utilization of 21-deoxycortisol and 21-deoxycortisone as “downstream” markers to detect 21OHD has been suggested, given that these molecules are produced when elevated 17OHP is diverted to other steroidogenic pathways [2,19,20]. 11-oxygenated androgens such as 11β-hydroxyandrostenedione and 11-ketotestosterone, derived from 21-deoxycortisol, have also been proposed as alternative markers [21]. Molecular testing may also improve the detection of 21OHD. For example, the use of PCR to analyze 12 common mutations associated with the *CYP21A2* gene in 39 previously identified cases of 21OHD yielded a reported genotype to phenotype concordance of 89.7% [22]. However, the validity of genetic testing as a screening tool continues to be debated, given the rarity of CAH; additionally, the sensitivity and cost-effectiveness of molecular testing has yet to be fully determined [5].

## 5. Conclusions

It remains vital to explore other options for improving screening for 21OHD. The combined rarity and potential severe clinical consequences of undiagnosed 21OHD call for greater precision in screening practices. Families experience significant stress and uncertainty when a diagnosis is ambiguous due to an unreliable screening method. This study and others highlight the limitations of 17OHP measurement as a newborn screening marker for 21OHD, despite numerous screening modifications. Future studies should continue to explore alternative approaches that do not rely exclusively on 17OHP to improve newborn screening accuracy for 21OHD.

## Figures and Tables

**Figure 1 IJNS-08-00057-f001:**
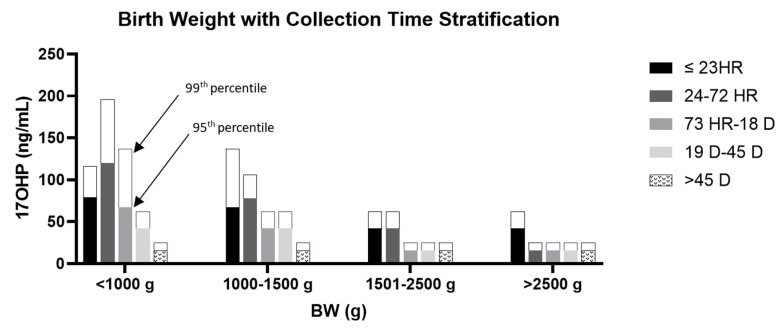
Distribution of 17OHP concentrations based on birth weight and collection time, with stratified 95th and 99th percentiles.

**Table 1 IJNS-08-00057-t001:** Sensitivity, specificity and PPV of 21OHD screening based on modified 17OHP percentile cutoffs.

**17OHP Cutoff at 99th Percentile**			
		**Presence of disease**	
		**21OHD ***	**Unaffected** **†**
**Test results**	**Screened positive**	53	355
	**Screened negative**	2	33,927
**Sensitivity**	0.963		
**Specificity**	0.989		
**PPV**	0.130		
**17OHP cutoff at 95th percentile**			
		**Presence of disease**	
		**21OHD ***	**Unaffected** **†**
**Test results**	**Screened positive**	54	1921
	**Screened negative**	1	32,361
**Sensitivity**	0.981		
**Specificity**	0.943		
**PPV**	0.027		

* 55 confirmed cases of 21OHD, regardless of whether they screened positive or negative, per proposed algorithm. †Unaffected cases are defined as the cohort of newborns, not diagnosed with 21OHD at the time of publication (n = 34,282).

## Data Availability

Not applicable.

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
