# Peer review of "Multiple 17-OHP Cutoff Co-Variates Fail to Improve 21-Hydroxylase Deficiency Screening Accuracy"

_2409-515X, 2022, doi:10.3390/ijns8040057_

Round 1

Reviewer 1 Report

The manuscript provided by Matharu, et al., provides insight into the potential for utilizing multiple co-variates to improve NBS performance metrics for 21OHD CAH. As NBS for this disease remains problematic in terms of both false positive and false negative results, this paper contributes to possible solutions in mitigating these types of results.

Relatively minor suggestions for improvement are below:

Lines 39-42: Elevations of 17OHP are described as being 'falsely elevated.' However, there are real and true elevations of 17OHP in this population - albeit not secondary to CAH. Consider rewording to indicate that the elevations are true and biological, but not caused by the underlying target disease.

Line 71-72: Be more clear that the n of 104,080 are specimens from unaffected newborns (at least I believe this to be true)

Line 76: Consider changing "unaffected specimens" to "Specimens from unaffected newborns" or "screen-negative results"

Line 100-104: Tie this back to explaining why you chose a time frame of 24-72 hours over 24-48 hours as it is not clear that this is explaining that upon first read.

Table and Figure 1: I found these a bit hard to follow - is there a way to choose more distinct colors (especially between 95 and 99% intervals) and put some additional spacing in the table?

Author Response

The manuscript provided by Matharu, et al., provides insight into the potential for utilizing multiple co-variates to improve NBS performance metrics for 21OHD CAH. As NBS for this disease remains problematic in terms of both false positive and false negative results, this paper contributes to possible solutions in mitigating these types of results.

Relatively minor suggestions for improvement are below:

Lines 39-42: Elevations of 17OHP are described as being 'falsely elevated.' However, there are real and true elevations of 17OHP in this population - albeit not secondary to CAH. Consider rewording to indicate that the elevations are true and biological, but not caused by the underlying target disease.

  • Thank you for this suggestion. The manuscript has been reworded. Line 39-42. “Reasons for a low PPV included elevated 17OHP levels in newborns shortly after delivery (especially during premature births) that are unassociated with 21OHD…

Line 71-72: Be more clear that the n of 104,080 are specimens from unaffected newborns (at least I believe this to be true)

  • We agree. The manuscript has been reworded.  Line 71-72. “The specimens were collected from unaffected newborns between January 1, 2019 and June 30, 2020 (n=104,080)

Line 76: Consider changing "unaffected specimens" to "Specimens from unaffected newborns" or "screen-negative results"

  • The manuscript has been reworded. Line 76. “Specimens from unaffected newborns were separated into two cohorts:…”

Line 100-104: Tie this back to explaining why you chose a time frame of 24-72 hours over 24-48 hours as it is not clear that this is explaining that upon first read.

  • We appreciate this suggestion. Clarity has been provided in lines 104-110 as to why 24-72 hour timeframe was used, instead of 24-48 hours.  Examination of 17OHP levels from unaffected newborns collected between 24-48 and 49-72 hours across our established BW categories did not yield significant differences between the two collection times within our particular data sets and therefore one grouping (24-72 hours) was made.

Table and Figure 1: I found these a bit hard to follow - is there a way to choose more distinct colors (especially between 95 and 99% intervals) and put some additional spacing in the table?

  • Thank you for this feedback. The figure has been updated to provide clarity, as suggested.

Reviewer 2 Report

A clearly written presentation of the influence of birth wight and age at sampling on the level of 17OHP in the neonatal screening for CAH. Such studies have been published before as pointed out by your group earlier, but this is a straightforward way to test the algorithm.

1.       Please describe what forms of CAH are the target of the study, SW, SV?

2.       What patients were missed with the adjusted recall levels? SW or SV?

3.       As far as I can calculate, the positive predictive value of the adjusted screening with recall at the 99th percentile was not 0,129 but 0,1299 equal to 0,130, according to the numbers in Table 1.

4.       What was the specificity, sensitivity and PPV of the screening program from which you chose the samples from unaffected infants (69,798 and the 34,282 samples) and how was that screening algorithm? Is it possible that some of these samples are from patients missed in the screening and diagnosed later? Maybe this possibility should be mentioned. Do you know that it is the very 55 true positive samples that have come out as positive in the data in Table 1? The reason for my question is the fact that the numbers add up so well without any “new” cases coming up with the improved screening algorithm. What would be the result if you look at the same groups with your unmodified recall levels (your “old” algorithms) with respect to specificity, sensitivity and PPV. It would be interesting to see this data in the Table 1 to visually illustrate the difference when weight and date of sampling is used in the present manner for the recall levels. As far as I understand from your publication in 2019 you are already using a birth weight and age at sampling algorithm in your screening programme. If so, it is not surprising that you get the same results in this study.

Author Response

A clearly written presentation of the influence of birth wight and age at sampling on the level of 17OHP in the neonatal screening for CAH. Such studies have been published before as pointed out by your group earlier, but this is a straightforward way to test the algorithm.

  1. Please describe what forms of CAH are the target of the study, SW, SV?

  • Please see lines 85-88 in manuscript. Stratification of cases by severity of disease, simple virilizing or salt wasting phenotypes, could not be performed due to reporting discrepancies of confirmed cases and lack of confirmatory molecular data.  

  1. What patients were missed with the adjusted recall levels? SW or SV?

  • Please see lines 85-88 in manuscript. Stratification of cases by severity of disease, simple virilizing or salt wasting phenotypes, could not be performed due to reporting discrepancies of confirmed cases and lack of confirmatory molecular data. 

  1. As far as I can calculate, the positive predictive value of the adjusted screening with recall at the 99th percentile was not 0,129 but 0,1299 equal to 0,130, according to the numbers in Table 1.

  • Thank you for noticing this slight miscalculation. Calculation was adjusted as recommended (0.130).

  1. What was the specificity, sensitivity and PPV of the screening program from which you chose the samples from unaffected infants (69,798 and the 34,282 samples) and how was that screening algorithm? Is it possible that some of these samples are from patients missed in the screening and diagnosed later? Maybe this possibility should be mentioned. Do you know that it is the very 55 true positive samples that have come out as positive in the data in Table 1? The reason for my question is the fact that the numbers add up so well without any “new” cases coming up with the improved screening algorithm. What would be the result if you look at the same groups with your unmodified recall levels (your “old” algorithms) with respect to specificity, sensitivity and PPV. It would be interesting to see this data in the Table 1 to visually illustrate the difference when weight and date of sampling is used in the present manner for the recall levels. As far as I understand from your publication in 2019 you are already using a birth weight and age at sampling algorithm in your screening programme. If so, it is not surprising that you get the same results in this study.

  • The current 17OHP screening cutoffs (based upon birthweight without stratification for time of collection) were applied to the unaffected newborn data set (January 1, 2020-June 30, 2020) and the 55 confirmed cases. A screening sensitivity of 90.9%, specificity of 96.4%, and PPV of 0.039 was achieved.  Five of the 55 true cases would have been “missed”.

It is difficult to compare the screening accuracy of the current cutoffs to the proposed cutoffs at the 95th and 99th percentiles.  The current cutoffs were not necessarily set at defined percentiles or standard deviations and do not use the same groupings for birthweight.    

Our goal with this manuscript was to demonstrate that 17OHP is not an ideal marker for CAH.  In order to identify all cases of CAH, the cutoff must be set very low, leading to a large number of false positives, even when stratifying by birthweight or collection time.  Future studies should continue to explore alternative approaches that do not rely exclusively on 17OHP to improve newborn screening accuracy for 21OHD.  

Reviewer 3 Report

Thank you for the opportunity to review this paper. It was well written, concise and again highlights the limitations of using immunoassay for screening for CAH. 

I would make the following suggestions

It would be useful to state whether SV-CAH and SW-CAH were screening targets and if the missed CASES using the co-variate 17OHP cut-offs were SW-CAH. 

It would be helpful to know the number of screening tests and false positive by BW categories as a n indication of the accuracy of percentile calculations in these groups.

Author Response

Thank you for the opportunity to review this paper. It was well written, concise and again highlights the limitations of using immunoassay for screening for CAH.

I would make the following suggestions

It would be useful to state whether SV-CAH and SW-CAH were screening targets and if the missed CASES using the co-variate 17OHP cut-offs were SW-CAH.

  • Please see lines 85-88 in manuscript. Stratification of cases by severity of disease, simple virilizing or salt wasting phenotypes, could not be performed due to reporting discrepancies of confirmed cases and lack of confirmatory molecular data. 

It would be helpful to know the number of screening tests and false positive by BW categories as a n indication of the accuracy of percentile calculations in these groups.

  • Thank you for the suggestion. The authors felt that incorporating this data into the manuscript would produce more questions and lead to confusion or a lack of clarity.  However, in future iterations of the screening algorithm, the authors will consider minor tweaks to the percentile cutoff applied to each BW/CT group.   

Round 2

Reviewer 2 Report

Thank you for the revision of the manuscript. The interesting part of the manuscript is Figure 1.

1. Please define clearly in the Materials and Methods under Specimens that the 55 positive cases comprise patients with SW or SV ( I suppose this is the case).

2. Table 1. Please clarify that the only patients you define as true positives are the 55 you have defined from start and that all infants inte the control group, which come out as positives in the screening are depicted as false positives (which may - after all not be true for all of them). We know well that the report back of false negative patients is not 100%.

3. Thus the numbers with sensitivity, specificity and PPV have to be interpreted with caution. Please mention this in the discussion.

4, The sentence on line 40 needs to be clarified.

Author Response

Thank you for the revision of the manuscript. The interesting part of the manuscript is Figure 1.

Thank you for your comments and review of our manuscript.

  1. Please define clearly in the Materials and Methods under Specimens that the 55 positive cases comprise patients with SW or SV ( I suppose this is the case).
  • Thank you for your comment. The manuscript has been updated to contain the following sentence: “Therefore, our pool of 55 confirmed cases was assumed to contain both phenotypes.” (Line 90-91)
  1. Table 1. Please clarify that the only patients you define as true positives are the 55 you have defined from start and that all infants in the control group, which come out as positives in the screening are depicted as false positives (which may - after all not be true for all of them). We know well that the report back of false negative patients is not 100%.
  • Thank you for this insight. The caption for Table 1 has been modified in the manuscript and includes the following clarifications:

*55 confirmed cases of 21OHD, regardless of whether they screened positive or negative, per proposed algorithm.

†Unaffected cases are defined as the cohort of newborns, not diagnosed with 21OHD at the time of publication (n=34,282). 

These two sentences have been denoted with an asterisk and dagger and correlate to each column in the table for reference (21OHD column vs unaffected column).

  1. Thus the numbers with sensitivity, specificity and PPV have to be interpreted with caution. Please mention this in the discussion.
  • The manuscript has been updated to include the following lines in the discussion:

“It must be noted that our study utilized a large cohort of newborns (34,282) presumptively unaffected with 21OHD as the control group.  Some of these newborns may have 21OHD but have not yet come to clinical attention at the time of publication (false negatives).  Therefore, the sensitivity, specificity and PPV as defined above should be interpreted with caution. “(Line 173-177).

  1. The sentence on line 40 needs to be clarified.
  • The manuscript sentences (line 39-41) have been modified as such: “Reasons for a low PPV include elevated 17OHP levels in newborns (especially premature infants) shortly after delivery that are not associated with a diagnosis of 21OHD….”

We hope this reads better and clarifies the observation that not all cases of elevated 17OHP are related to 21OHD.